# In Vitro Multiplication of *Lophostemon suaveolens* (Sol.ex Gaertn.) Peter G.Wilson & J.T. Waterh): Peatland Tree Species for Rehabilitation

**Asri Insiana Putri \***, **Noor Khomsah Kartikawati, Arif Nirsatmanto**, **Sri Sunarti, Liliek Haryjanto,**
**Toni Herawan, Purwanto Budi Santosa, Reni Setyo Wahyuningtyas, Fajar Lestari and Anto Rimbawanto**

National Research and Innovation Agency (BRIN) Republic of Indonesia, Jl. Ir. H. Juanda No. 18 Paledang,
Kota Bogor 16122, Jawa Barat, Indonesia
* Correspondence: asri008@brin.go.id; Tel.: +62-81559728514

**Abstract:** Peatlands in Indonesia are one of the world's largest carbon sinks, helping to regulate greenhouse gas emissions and global climate change. *Lophostemon suaveolens* is a relatively unexplored plant found in Papua's endemic peat ecosystem that grows well in wet areas with low fertility. It is geographically dispersed and has the potential for peatland rehabilitation. Seed is one of materials for the reproduction of *L. suaveolens*. However, the difficulty in seed collection and the limitation in seed production has become a current problem for its cultivation. Seed multiplication by using an in vitro method would be one of the mechanisms to overcome the problem. We present an efficient and reproducible protocol for in vitro multiplication of plantlets using nodal segments and shoot apices collected from plantlets. After 3 months of the culture initiation stage, the elongated axillary shoots were separated from the clumps and further multiplied using Murashige and Skoog (MS) media supplemented with (1) BAP (0.5 mL/L) as single PGR, (2) NAA (0.1 mL/L) as a single PGR, and (3) a combination of two types of PGR BAP (0.5 mL/L) and NAA (0.1 mL/L). Up to an incubation period of 6 months, the efficiency of leaf axillary shoot propagation was determined by counting the number of nodule multiplication coefficient (NMC), shoot length, root length, and number of leaves (six consecutive subcultures). The higher the NMC, the higher the plantlets obtained, increasing shoot regeneration from nodules physiologically increasing evapotranspiration in vitro. The highest of NMC (8.4) was observed in MS medium with a combination of 0.5 mL/L BAP and 0.1 mL/L NAA (double PGRs), with the longest shoots (5.91 cm), the longest root length (8.83 cm), and the most leaves (32). When a combination of BAP and NAA were used simultaneously, the plantlets during acclimatization were the highest survived. It was concluded that MS in combination with 0.5 mL/L BAP and 0.1 mL/L NAA is the most appropriate protocol for the success of in vitro multiplication of *L. suaveolens*. This is the first report of *L. suaveolens* in vitro multiplication, and the protocol could be used to propagate this peatland species on a large scale. The authors acknowledge the limitations of the experimental work and recommend further work to increase the sample size and complete the field-testing phase to help verify the initial findings presented in this paper.

**Keywords:** acclimatization; axillary shoots; BAP; NAA; propagation

## 1. Introduction

Indonesia has the world's largest tropical peatland [1]. It is estimated that tropical peat land retains approximately 40% of terrestrial carbon [2]. Traditional slash and burn practices for agricultural and plantation purposes have resulted in large peat fires that released a large amount of carbon into the atmosphere [3], potentially affecting atmospheric composition and the climate system. In 2015, the worst peat fires since 1997 ravaged Sumatera, Kalimantan, and Papua. The sensitivity of peatland degradation varies depending on geographic location. Zhang [4] reported on the sensitivity of peat degradation via remote sensing observation of trench ecology in the Zoige peatland, East Tibetan plateau. Because

peat soils decompose more strongly at higher elevations, Alpine peat is more sensitive to disturbances associated with the groundwater table. Valjarevic et al. [5] used remote sensing, topographic maps, geographic information system (GIS) analysis, and official data from cadastral and census books to reconstruct forest conditions, including degraded land, in Toplica, Republic of Serbia, over the last 60 years. The combination of remote sensing techniques and micro-techniques can be a mutually reinforcing option for land rehabilitation. The combination of remote sensing techniques and micro-techniques can be a mutually reinforcing alternative for the rehabilitation of degraded land.

Peatland rehabilitation aims to restore former peatland ecosystem processes, productivity, and services, but it does not imply restoring pre-existing biotic integrity in terms of species composition, community structure, and ecosystem functions [6]. Peatland restoration, also known as the process of restoring peatlands, refers to the process of restoring peatland ecosystems.

Peat restoration, also known as peatland restoration process, refers to the process of restoring peatland ecosystems. Peatland restoration, or the process of returning degraded or degraded peatlands to their natural state, takes this a step further [7]. The global policy for the rehabilitation of marginal lands is focused on the concept of tropical forest and landscape restoration. Intensification techniques are the focus of ecology to improve crop quality in degraded landscape areas [8]. Unlike well-established studies on ecological restoration of isolated peatlands [7], tropical peatland restoration is still in its early stages [3,9,10]. This research supports the vegetative propagation of quality plants for land rehabilitation. The basic operational model for the purpose of reforestation using in vitro propagation techniques has been carried out in Brazil, mainly for 15 tree species, 10 woody trees (Couroupita guianensis Aubl., Tabebuia heptaphylla (Vell.) Toledo, Tabebuia impetiginosa (Mart. ex DC.) Standl., Tabebuia roseoalba (Ridl.) Sandwith, Vochysia haenkeana (Spreng.) Mart., Vitex montevidensis Cham., Copaifera coriacea Mart., Spondias tuberosa Arruda, Schinus terebinthifolia Raddi, and Talisia esculenta (A.) Radlk.-Hil), and 5 palm trees (Syagrus coronate (Mart.) Becc., Attalea oleifera Barb. Rodr., Elaeis guineensis Jacq., Colubrina glandulosa Perk., and Astrocaryum vulgar Mart.) [8].

According to the ministerial regulation, one of the plant criteria for peat ecosystem rehabilitation was to prioritize native species and land suitability. *Lophostemon suaveolens*, a plant in the Myrtaceae family, was found in the peat ecosystem. *L. suaveolens* is a tree or shrub with terminal buds covered in scales, stems, and petioles that exude a milky resin when cut. On mature branches, the leaves alternate and the inflorescences are axillary and cymose [10]. *L. suaveolens* is the dominant tree species in the Merauke area, and it is widely used by the community for building materials, firewood, and a variety of other purposes [11,12]. Forest rehabilitation is the process of restoring degraded forest structures in order to generate forest products and productivity, but it does not always include all plant and animal species that are expected to exist in a given location. A large number of long-lasting seeds are required for large-scale restoration. Seedling propagation must be efficient in order to meet demand on a large scale. Seed is one of the materials for the reproduction of *L. suaveolens*. However, the difficulty in seed collection and the limitation in seed production becomes a current problem for its cultivation. Seedlings of *Lophostemon* sp. cannot be propagated naturally in sufficient numbers. Micropropagation is an alternative method of vegetative reproduction to traditional methods. In forestry, in vitro propagation techniques are useful for mass reproduction of superior trees found in nature, and axillary shoot proliferation has been found to be the most successful in achieving plantlet regeneration from mature forest trees [13].

One technique for vegetative propagation is multiplication in vitro culture. In vitro culture is a propagation technique that isolates certain plant parts and then stimulates them to multiplication and rooting to form new plants by modifying the growing media and environment suitable for plants. In vitro culture techniques are able to produce large amounts of planting material in a relatively short time, are uniform, free of pathogens, and do not depend on the season. Multiplication in in vitro culture, growth, and development

of explants is strongly influenced by the type of base media and plant growth regulators. To increase explant propagation, growth regulators in the form of auxin and cytokinin groups are generally added to the growing media. This growth regulator stimulates the process of growth and development of plant tissue [14]. The auxin-cytokinin ratio determines the type of culture established or regeneration type. 1-Naphthaleneacetic acid (NAA) is classified as an auxin, and 6-Benzylaminopurine (BAP) is classified as a cytokinin based on their primary function in in vitro and ex vitro propagation [15]. NAA, which is synthesized in the shoot apex of young leaves and transported basipetally to the roots [16], is essential for maintaining apical dominance and cell elongation. Its main effects include rooting, stimulation, and inhibition of axillary bud outgrowth [17]. Cytokinin regulates meristem activity [18], plant shape determination, side condition adaptation, and environmental responses. NAA and BAP act either antagonistically or synergistically to control developmental processes, such as meristem formation and maintenance [19]. BAP and NAA are a type of PGR, most often used in shoot propagation because these are stable, resistant to degradation, and readily available. Murashige and Skoog (MS) media are the basic media that are generally used for the propagation of various types of plants. This basic medium is rich in minerals that stimulate organogenesis. To increase the propagation of explants, growth regulators in the form of auxin and cytokinin groups are usually added to the growing media. This growth regulator stimulates the process of growth and development of plant tissue [20].

In vitro multiplication has emerged as a successful and effective technique for mass propagation of species suitable for peatland rehabilitation. The lack of efficient propagation methods, combined with the high growth potential in these marginal peatlands, highlights the importance of germplasm and the development of in vitro propagation protocols. The purpose of this study was to observe the addition of exogenous BAP and NAA, which were thought to increase the multiplication coefficient, increase shoot elongation, root elongation, and number of leaves on *L. suaveolens* to maintain the availability of stronger vegetative propagation seeds for rehabilitation on marginal land. Stable nursery architecture will promote optimal growth in nursery acclimatization. The effect of NAA and BAP on the in vitro multiplication of *L. suaveolens* has not been studied in depth.

This research has a strategic value to obtain techniques for providing plantlets or seedlings on marginal lands that are difficult to manage. In vitro enrichment of PGRs has the potential to strengthen seedlings resulting from more efficient vegetative multiplication on a large scale. The weakness of this study is that it does not fully display the entire micropropagation process, starting from the preparation of explant material from trees in nature to field trials of planting seedlings resulting from in vitro multiplication. However, it can actually focus more on displaying research results on the most important part of in vitro multiplication for *L. suaveolens*, which has not been reported by other studies

## 2. Materials and Methods

The research was carried out at the tissue culture laboratory, greenhouse, and nursery at the Center for Forest Biotechnology and Tree Improvement in Yogyakarta, Indonesia (7°40′20″ S and 110°23′30″ E), 600 m above sea level, with the highest temperature being 32 degrees Celsius and the lowest temperature being 18 degrees Celsius, and a mean humidity of 95%. Researchers observed for 13 months in the laboratory and 14 months in the greenhouse from January 2018 to March 2021. Genetic material source and newly formed shoots of plantlet were taken from PT. Arara Abadi tissue culture laboratory at Riau, Sumatera, Indonesia. The equipment used in this study met tissue culture laboratory standards.

## 3. Multiplication of *L. suaveolens*

The goal of this phase was to produce more *L. suaveolens* propagules from shoots. Micro cutting techniques and repeated subcultures multiplied the number of propagules until the desired number of plants was obtained. The treatment composition was transferred into several new culture media after the nodule of the axenic plantlets was separated as an

explant. This study's in vitro multiplication of *L. suaveolens* is a follow-up to preliminary observations on the initiation of axillary buds on various basic media and PGRs (data not published). Several modifications of explants disinfection method by [21] were used in this study to obtain aseptic explants. Disinfection of explants was continued in the LAF by soaking and shaking the explants for 40 min in 15% (*v/v*) antimicrobial compound biocide isothiazolone (BI) and a few drops of Tween[80] for 40 min. The final step of disinfection was to soak and shake the explants for one minute in 70% ethanol. After every disinfection, explants were rinsed with sterile distilled water.

Aseptic explants were cultured in glass tubes (25 mm 150 mm) with 25 mL of MS medium supplemented with single PGR treatment (one kind of BAP 0.5 mL/L and one kind of NAA 0.1 mL/L) and double PGRs treatment (combination of 0.5 mL/L BAP and 0.1 mL/L NAA). Before being gelled in 8 g/L of Phyto agar (King of Fiber, Jakarta, Indonesia) for multiplication and 6 g/L of Phyto agar for rooting, each basal medium contained vitamin and was supplemented with 30 g/L sugar (Sugar Group Companies, Jakarta, Indonesia). Before autoclaving at 121 °C, the pH of the medium was adjusted to 5.8 by adding a few drops of 1 N hydrochloric acid 32% (Merck 100313) or sodium hydroxide (Merck 1310-73-2), and all steps were repeated three times to ensure reproducibility in media preparation. The experimental setup was kept in a laboratory at 25 °C and 60% relative humidity under white fluorescent light at 40 mol/m$^2$/s intensity with a 16 h light and 8 h dark cycle.

Each PGR and PGRs treatment at this multiplication stage was tested in 20 culture replicates, each culture containing one explant consisting of one nodule. The multiplication potential of explants was expressed as nodule multiplication coefficient (NMC), shoot length, root length, and number of leaves of *L. suaveolens* in vitro, which were determined after 6 months of incubation. The following shows how NMC is defined:

$$\text{NMC} = \text{VA} \times \text{AN} \times (\text{NPT})^{-1} \times 100\% \tag{1}$$

NMC: nodule multiplication coefficient, VE: viable explant, AN: average of nodules/explant, NPT: number of plantlets transplanted.

*Rooting and Acclimatization of L. suaveolens*

After 6 months of incubation and rooting regeneration, another 4 months without subculture was performed to ensure that the plantlets were healthy with well-developed roots in vitro prior to acclimatization in the green house. The plantlets were washed under running tap water, soaked in fungicide (Mankozeb 80%) for several minutes, and transplanted into a polybag containing a mixture of regosol topsoil (sandy soil) and compost from bamboo leaf litter in a 1:1 ratio. Acclimatization of plantlets was done in a greenhouse with 30–40% natural light and watered once a day for 6 months. Total 50 plantlets were individually transplanted in a poly bag with a plastic cover in this experiment. There was no control in acclimatization, just make sure the best rooting in vitro could survive in soil medium and green house conditions.

Acclimatization was carried out carefully by removing the plantlets from the glass tube, and the roots were washed with running tap water to remove the adhering agar and soaked in fungicide (Mankozeb 80%) for several minutes. Plantlets were transferred to polybags containing a mixture of regosol topsoil and compost from bamboo leaf litter in 1:1 ratio. Acclimatization was carried out in a greenhouse with 30–40% natural light and watered once a day for 6 months. In this experiment, 50 plantlets were transplanted one by one into polybags with plastic covers without control. The growth of plantlets was observed in the nursery after 8 months. The following formula is used to calculate the percentage of plantlets that survive after planting

Plantlets growth was observed at the nursery after 8 months. The following formula was used to calculate the percentage of plantlets that survived after being planted:

$$\text{SP} = \text{NPS} \times (\text{NPT})^{-1} \times 100\% \tag{2}$$

SP: survived plantlets, NSP: number of survival plantlets, NPT: number of plantlets transplanted.

A completely randomized design (CRD) with 3 replicates and at least 20 explants per replicate was used in the experiment. The data was analyzed using ANOVA, and the mean values were separated using Duncan's multiple range test (DMRT) [22] at a significance level of =0.05. The statistical package SPSS (Version 24) was used for analysis.

## 4. Results

### 4.1. Multiplication of L. suaveolens

Continuous subculture on the same medium resulted in multiple shoot regeneration in explant nodules. Figure 1 depicts the multiplication of *L. suaveolens* in vitro; these plantlets were separated and transferred to new media for the rooting stage. The separation of plantlets into several new cultures then elongate the shoots again and produce multiple nodules again. This multiplication capability is an advantage for the purpose of mass seed production because it does not occur in every species.

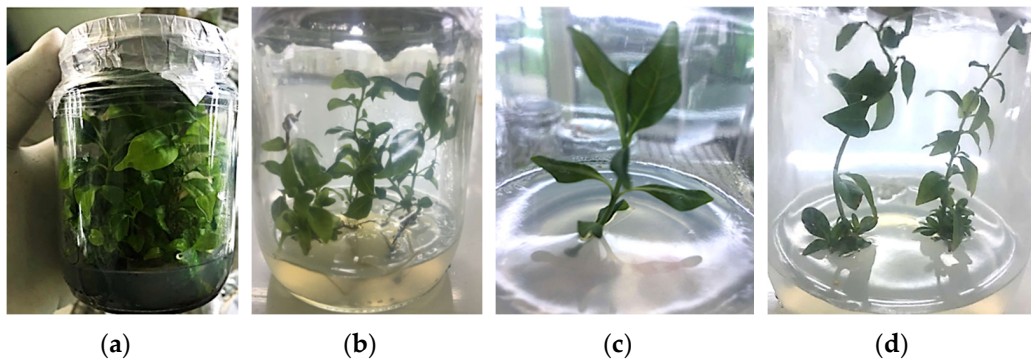

(**a**)      (**b**)      (**c**)      (**d**)

**Figure 1.** In vitro multiplication of *L. suaveolens* plantlets as the source of explants from axenic culture (**a**), separation of the axenic plantlets into several new culture media (**b**), explants of multiple nodules from shoot elongation (**c**), shoot elongation of explants after 6 months sub-culture (**d**), (Credit: Putri 2018).

Figure 2 shows the sprouting ability in multiplication of *L. suaveolens* in vitro, regeneration in media with single PGRs BAP, regeneration in media with single PGRs NAA, and regeneration in media with BAP and NAA after 36 subcultures.

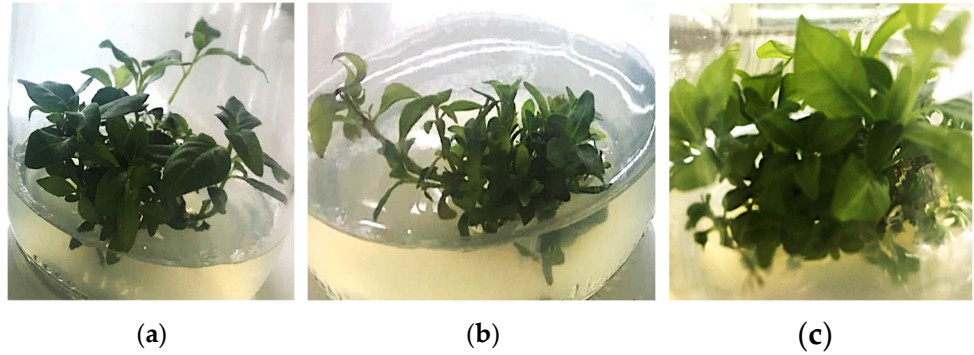

(**a**)      (**b**)      (**c**)

**Figure 2.** Sprouting ability in in vitro multiplication of *L. suaveolens* regeneration in media with BAP (**a**), regeneration in media with NAA (**b**), regeneration in media with combination of BAP and NAA (**c**) (Credit: Putri 2018).

Table 1 shows the effect of single and multiple PGRs on the multiplication parameters of *L. suaveolens* after 6 months of in vitro incubation. The number of nodule multiplication coefficients (NMC) are significantly different in each treatment, as well as for shoot elongation. Root elongation was not significantly different between single hormone treatments

but significantly different between single hormone and double hormone combinations. A combination of double PGR additions were higher than single PGRs in NMC ($8.4 \pm 0.13$), shoot length ($5.91 \pm 0.17$ cm) and root length ($8.83 \pm 0.69$ cm). The multiplication stage in this study obtained 100% axenic cultures, thus all cultures could be used to calculate the multiplication coefficient. According to the observations, all shoots are from axillary, not adventitious shoots. The ability to germinate after being transferred to a new medium is indeed different; some are single shoots and some can form multiple shoots. This ability is very advantageous at the multiplication stage. During the observation that callus growth did not occur, there may have to be additional exogenous hormones to grow callus.

**Table 1.** Effect of single and double PGRs on *L. suaveolens* multiplication parameter after 6 months incubation in vitro. Each value is a mean of 20 replicates with standard error (Mean $\pm$ S.E). Mean with different letters (a,b,c) in each column are significantly different at 0.05 probability level by the Duncan multiple range test. The results of ANOVA showed significant differences ($p \leq 0.05$) between the concentration of NAA, BAP, NAA, and BAP tested for NMC, shoot length, and root length.

| PGRs | Nodule Multiplication Coefficients (NMC) | Shoot Length (cm) | Root Length (cm) | Leave Number |
|---|---|---|---|---|
| BAP | $6.0 \pm 0.07$ [b] | $4.72 \pm 0.13$ [b] | $5.03 \pm 0.11$ [a] | $30 \pm 0.01$ [a] |
| NAA | $5.1 \pm 0.09$ [a] | $2.21 \pm 0.09$ [a] | $5.10 \pm 0.09$ [a] | $29 \pm 0.12$ [a] |
| BAP and NAA | $8.4 \pm 0.13$ [c] | $5.91 \pm 0.17$ [c] | $8.83 \pm 0.69$ [b] | $32 \pm 0.07$ [a] |

Nodule multiplication efficiency of *L. suaveolens* in vitro for 6 months incubation is the initial multiplication time of the average of several plantlets. On a mass scale, this multiplication is carried out on thousands of plantlets continuously. During the first 6 months, the highest yield was 8.4 nodules/explant and each nodule produced 1 to 3 shoots, so that approximately 24 new shoots were obtained per explant to become 24 new plantlets. As a comparison, the vegetative propagation of *L. suaveolens* by cuttings conducted by Kartikaningtyas et al. [23] resulted in 3 new shoots from one stem cutting without incubation time restrictions, and root formation did not occur. Seedlings material is obtained from seeds, which are very limited in availability for large areas of land rehabilitation, so the information in this study is very important to be able to continue research on a mass scale.

*4.2. Rooting and Acclimatization of L. suaveolens*

Figure 3 depicts root elongation after a 10-month subculture (6 months for multiplication and rooting regeneration, and another 4 months to ensure that the plantlets were healthy, with well-developed roots in vitro prior to acclimatization), with exogenous BAP, exogenous NAA, and double PGRs.

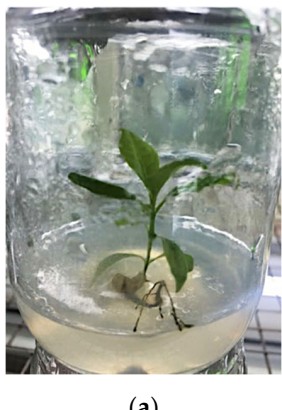
**(a)**

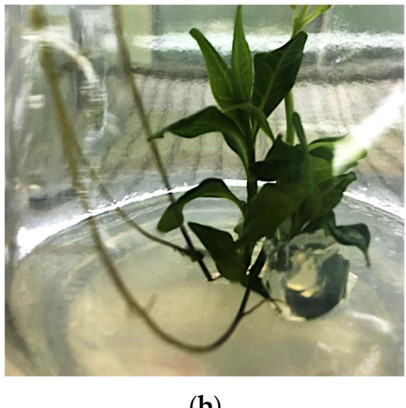
**(b)**

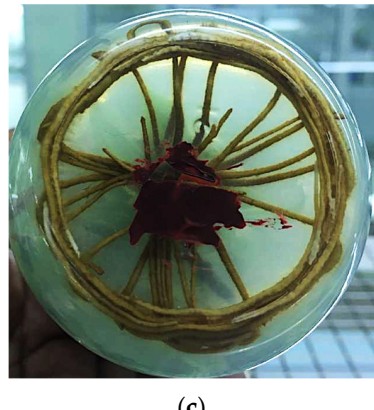
**(c)**

**Figure 3.** Root lengthening of *L. suaveolens* after 10 months incubation (one subculture per month) with BAP (**a**), NAA (**b**), and double PGRs (**c**), (Credit: Putri 2018).

Figure 4 depicts the transition of a healthy *L. suaveolens* from in vitro regeneration to ex vitro development in a green house. After hardening, the rooted shoots were transferred to paper cups (Figure 4d). The leaf color changed to dark green 2 weeks after the plastic cover was removed, and a wax coat and villi were observed on the leaf surface; the stems were strong (Figure 4e). Plantlets were transplanted to pots after 6 months (Figure 4f), with survival rates of 97%, 92%, and 94% for double PGRs, BAP, and NAA treatments, respectively (Figure 5).

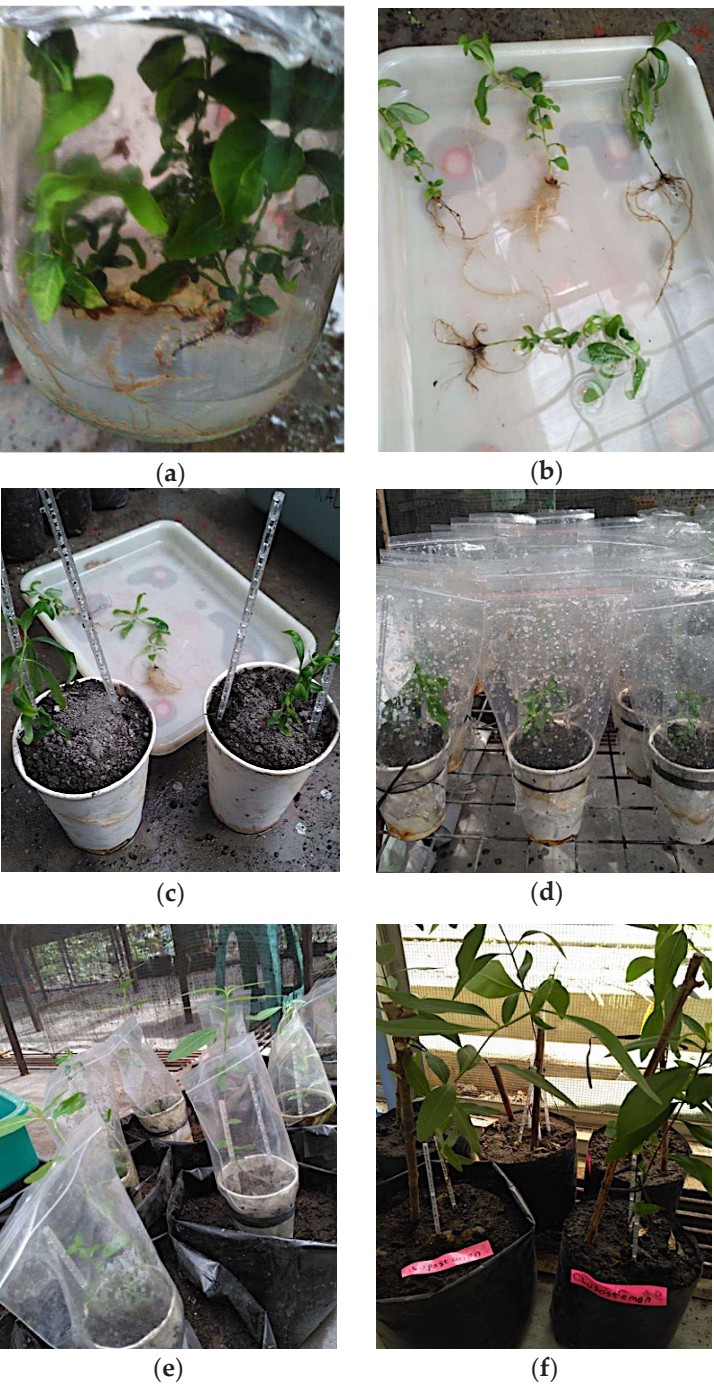

(a)    (b)

(c)    (d)

(e)    (f)

**Figure 4.** Acclimatization of *L. suaveolens* in greenhouse healthy plantlet from in vitro regeneration (**a**), in fungicide solution (**b**), and after being transplanted to the media with individual plastic cover (**c,d**). Tissue culture seedlings grow well after opening the plastic cover at 4 weeks of incubation (**e,f**), (Credit: Putri 2018).

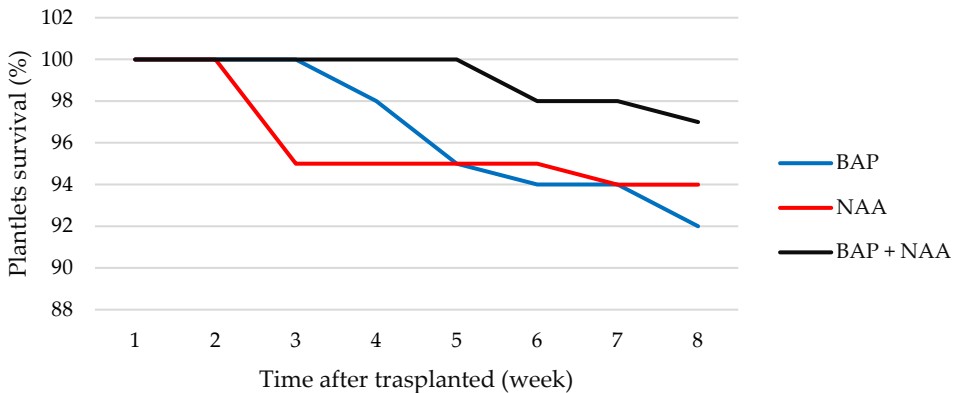

**Figure 5.** The percentage of *L. suaveolens* plantlets survival with PGRs treatments 8 weeks after transplanting. Each value is a mean of 20 replicates, significantly different from each other at 0.05 probability level by the Duncan multiple range test. The results of ANOVA showed a significant difference ($p \leq 0.05$) between NAA, BAP, NAA, and BAP treatment for plantlets survival.

The percentage of plantlets that survived after single and double PGR treatments were measured 8 weeks after acclimatization (Figure 5). The plants were then kept in a greenhouse for 8 months. There were no deaths in any of the treatments, and the combination of BAP and NAA treatment was the best. Plantlets care is currently limited to watering and no fertilizing.

Figure 6 depicts the post-acclimatization shoot height of *L. suaveolens* seedlings treated with PGRs. The plants were then kept in a greenhouse and grew normally, with no noticeable morphological differences from stock plants. The regenerated plants showed 100% survival after 8 months of growing in the green house, with an average height of $43 \pm 0.34$ cm for double PGRs, $37 \pm 0.21$ cm for BAP, and $38 \pm 0.53$ cm for NAA treatments.

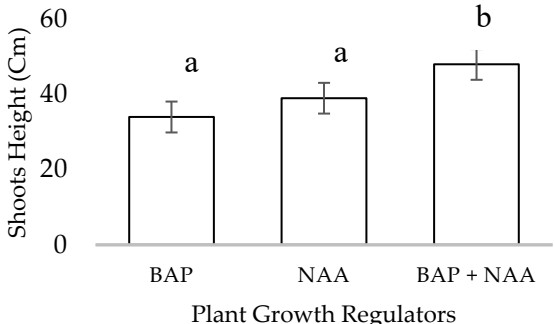

**Figure 6.** The shoot height of *L. suaveolens* plantlets in the green house. Each value is a mean of 20 replicates with standard error (Mean $\pm$ S.E). Mean with different letters (a,b) are significantly different at 0.05 probability level by by Duncan multiple range test. The results of ANOVA showed a significant difference ($p \leq 0.05$) between NAA, BAP, NAA, and BAP treatment for shoot height.

## 5. Discussion

### 5.1. Disinfection Culture Condition

The success of the in vitro multiplication study of *L. suaveolens* is highly dependent on the effectiveness of the disinfection stage to obtain axenic culture and can maintain axenic conditions continuously until new plantlets are obtained so that the subsequent multiplication regeneration process is not disturbed by contamination. The results of this study showed that the disinfection conditions of *L. suaveolens* culture could be maintained until the plantlets were obtained in each treatment in vitro up to 100%.

In this study, we used Biocide Isothiazolones (BI) as a disinfectant. BI utilizes a two-step mechanism involving rapid inhibition (minutes) of growth and metabolism, followed by irreversible cell damage, resulting in loss of viability (hours). Cells are inhibited

by disruption of the metabolic pathways involving dehydrogenase enzymes. Critical physiological functions are rapidly inhibited in microbes, including growth, respiration (oxygen consumption), and energy generation (ATP synthesis). Axenic culture under these conditions was able to maintain explant regeneration in an in vitro environment. The BI unique mechanism produces a broad spectrum of activity, which is expected to increase the endophytic contaminant disinfection ability, low usage rate, and difficulty in achieving resistance [24].

The disinfectant efficiency can be achieved through optimized disinfectant concentrations, as well as exposure periods [25–27]. Although better disinfection can be obtained with high disinfectants concentrations with longer exposure, explant viability can be negatively affected by disinfectants under these conditions, resulting in yellow dehydrated explants, along with low viability [26]. The desirable disinfection procedures should be proposed in a cheap, simple, efficient, and environmentally friendly way for eliminating the endogenous and surface contaminations [28].

*5.2. Multiplication of L. suaveolens*

Our preliminary research revealed that MS media produced the best response to the initiation of *L. suaveolens* axillary buds when compared to WPM (data not published). At this early stage, no PGRs were added to test the ability of endogenous PGRs to regenerate in vitro. After 8 weeks of incubation, there was a decrease in growth, preventing the multiplication stage. Exogenous PGRs were used as an explant source in this study, and the results were as expected. Growth regulators are organic substances that plants require in small amounts to regulate plant growth [29].

The results showed that the axillary shoots of *L. suaveolens* were successfully used as a source of explants for in vitro multiplication. The increase in the number of nodules formed as a result of shoot elongation indicates the effectiveness of the hormone treatment in this study. Moreover, the formation of multiple shoots per nodule further increase the effectiveness and efficiency of *L. suaveolens* multiplication. In a previous study, [16] used species that can only form multiple shoots to calculate the shoot multiplication coefficient; however, *L. suaveolens* can form multiple nodules from elongation in one shoot, as well as multiple shoots. The explants were placed in new culture media and incubated in a sterile environment (Figure 1d).

At the multiplication stage, the ability to form explant shoots on single PGR media was lower than on double PGR media subcultures. Figure 2 depicts the high sprouting ability of *L. suaveolens* at the multiplication stage. The concentrations and combinations of PGRs and endogenous hormones in the culture medium influenced the growth qualities of micro-propagules [17,18]. Although each species responds differently to the combination of endogenous and exogenous PGRs, BAP and NAA supplementation may cause hormonal imbalance in plants [19]. Our findings suggest that the response of *L. suaveolens* plantlets to BAP and NAA may differ between species. The ability of *L. suaveolens* to form multiple shoots (Figure 2a–c) demonstrates that this study was successful in revealing this species' high in vitro multiplication potential to provide plants for peat rehabilitation. This is the first time this technique for *L. suaveolen* has been studied.

According to George [30,31], hormones can be found naturally in plant metabolic systems (endogenous) or synthetically (exogenously). They help to promote cell differentiation, growth, and division. BAP and NAA are the two most common classes of PGRs used in in vitro growth techniques. The type of culture specified, or regeneration type, is determined by a comparison of BAP and NAA [32]. In terms of plant sensitivity to PGRs, the ratio varies by genus and species. The comparison of BAP and NAA is a determinant in meristem formation, and the balance of PGRs has been shown to be capable of stimulating the type of shoot formation. Endogenous PGR levels in plants influence exogenous PGR intake and also depend on the plant species.

The auxin-cytokine ratio in vitro multiplication varies depending on the type of plant used; this is critical to the success of in vitro plant regeneration. Single shoots can be propa-

gated from the shoot lengthening nodules, while multiple shoots can be separated. More research is needed to determine the ability of *L. suaveolens* to respond to a combination of PGR types and concentrations on plantlet growth. PGRs influence the genetic mechanism and regulate the differentiation and multiplication processes through differential gene action and cellular events, such as replication, transcription, and translation [33]. Plantlet regeneration in a variety of tree species, including *Eucalyptus nitens* [34], *Paulownia tomentosa* [35] and *Himalayan poplar* [36], has been reported using various explants and modulating auxin-cytokinin balances. BA (2.2 M) in combination with NAA (0.1 M) increased axillary budding and shoot multiplication in *Acacia mangium* and *Acacia mangium* × *Acacia auriculiformis* [37].

The addition of single BAP, single NAA, and double PGRs all resulted in significant differences in nodule multiplication coefficients and average shoot length, but with single PGRs being lower than double PGRs (Table 1). For an average root length, double PGRs have a significant difference compared to single PGRs, but there is no significant difference between the three treatments on average leave number. For large-scale propagation, the multiplication coefficient number of shoots regenerated per explant/nodule of *L. suaveolens* is the most important factor to consider. The cumulative effect of PGR combinations on the percentage of viable explants, average nodule per explant, NMC, average shoot elongation, and number of leaves was observed and measured. The combination of BAP and NAA has the potential to provide a better effect than single PGR, as shown in Table 1. BAP is the most effective growth hormone for shoot development in general. It works by inhibiting apical dominance of shoot induction and shoot bud formation.

According to several studies, media supplemented with BAP and NAA as double PGR is beneficial for shoot multiplication. One study found that MS media supplemented with 2 mg/L BA and 2 mg/L NAA yielded the best results. Another study discovered that 2 mg/L BAP resulted in the most shoot multiplications [38]. Using 2.22-M BAP, the median number of shoots produced Myrtaceae was determined. In some clones, this dose resulted in extremely high shoot numbers; explants frequently produced extensive callus and multiple short shoots. PGRs play an important role in cytodifferentiation and morphogenesis, regulating the complex processes of cell proliferation, xylogenetic differentiation, and root-shoot differentiation [39].

*5.3. Rooting and Acclimatization of L. suaveolens*

The addition of exogenous PGR resulted in greater root elongation after 6 months of incubation (Figure 3). There was no discernible difference in the number of leaves among treatments. After 6 months of in vitro incubation, exogenous PGR had no effect on leaf regeneration. To facilitate acclimatization, the ability to successfully root regenerated shoots in vitro is required. This study looked at the role of different PGR compositions in root induction in sub-cultured plants for 4 months. Full strength solid MS medium containing 0.5 mL/L BAP and 0.1 mL/L NAA (double PGRs) achieved the best in vitro rooting of individual shoots among the PGRs concentration treatments (Table 1). The average root length was greater in the double PGRs treatment than in the single PGR treatments (8.83 0.69 cm). Healthy plantlets with well-developed roots were transferred from the culture tube after 10 months of in vitro rooting regeneration for ex vitro acclimatization in a green house. In this study, 100% of the micro-cutting explants in the multiple PGR treatments were healthy, compared to 72% in the single BAP treatment and 86% in the single NAA treatment. The high mortality during or after transfer from in vitro to ex vitro conditions is a limitation in the large-scale application of tissue culture techniques.

The results of this study show that this species is highly resistant to abiotic stress (changes in temperature, light intensity, and humidity) and biotic stress (changes in the axenic culture environment to an abundant microbial environment in soil media) during acclimatization. Because plants lacked sufficient resistance to the soil microflora, their sudden exposure (particularly the root system) to microbial communities present in the soil was the leading cause of plant mortality during acclimatization [40]. Acclimatization is essential for improving plantlet survival and obtaining ready-to-plant seedlings. Direct

planting of seedlings resulting from plant tissue culture into the field was not possible due to the high mortality rate of plants. In vitro control of humidity, temperature, light, and nutrient availability necessitates ex vivo adjustments. Acclimatization to the various biotic environments in soil media requires special care as well.

In terms of seedling survival of *L. suaveolens*, the combination of BAP and NAA produced the best results (Figure 5). All seedlings can live healthy lives until the fifth week, but there is a slight decrease due to the death of some seedlings. Previous research has discovered that a variety of factors can cause seedling death during acclimatization. Because the cultured plants have nonfunctional stomata, a weak root system, and a poorly developed cuticle, transferring micro shoots to ex vitro conditions results in death [15]. Plantlets that received an excessive amount of PGRs developed morphological and anatomical abnormalities and were dubbed vitrified plants or hyperhydrate plants. Micro-propagated plantlets, due to their physiology and anatomy, must gradually adapt to the greenhouse or field environment [41]. However, plant mortality occurred in all treatments after the eighth week. At this point, plantlet care is limited to watering and no fertilization; this condition also demonstrates that a plant that can survive in waterlogged areas can also thrive in nutrient-deficient soils.

*L. suaveolens* has the potential to be one of the species used for revegetation and restoring peatland ecosystems that adapted to wetlands and have economic value in cultivation functions. Given the importance of peatland rehabilitation, the scarcity of seedlings for millions of hectares of degraded peatlands, and the lack of protocols for *L. suaveolens* regeneration in vitro, this study attempted to develop an in vitro propagation protocol that was evaluated for faster propagation. The contribution of this research to science includes the refinement or improvement of protocol-specific in vitro multiplication techniques for the addition of PGRs NAA and BAP for *L. suaveolens* species and research originality or novelty of the species, which has never been done before.

In vitro multiplication is the most important step in tissue culture propagation; the technique has never been used before for *L. suaveolens*. Revegetation will ensure the sustainability of the ecosystem, protect the peat from erosion, and provide economic value to local communities living near the peatland ecosystem (Indonesian Water Portal, 2021) [42]. Myrtacea is an alternative species that has been widely developed for land rehabilitation. Within one genus, it is possible to have close identification physiologically and morphologically, so it is possible to do in vitro multiplication, as is the case with *L. suaveolens*. The technique can be used as a support for environmental conditions that are degraded in a sustainable manner.

In general, in vitro plant multiplication techniques have been applied to the agricultural and forestry industries on a large scale, but for the purposes of planting production, they are not widely used for conservation purposes. The cost of maintaining the environment can be emphasized by mastering effective and efficient biotechnology propagation, as well as mastering the multiplication technique in this study.

## 6. Implication

A sufficient supply of long-lasting seedlings is required for large-scale restoration. Effective seedling propagation is required to meet demand on a large scale. When wild plants are scarce due to the time lapse since the last mass flowering of many forest plant species, tissue culture of *L. suaveolens* can provide an ongoing supply of plant seedlings. Tissue culture techniques have been developed to enable large-scale production of trees while also making these techniques more cost-effective. The cost of producing wildlings grows exponentially over time, whereas the cost of producing cuttings grows more slowly.

The highest NMC value is 8.4; this indicates that this species has the potential to produce eight nodules per shoot. Multiple shoots will produce a minimum of 10 shoots per plantlet, then a plantlet will produce 80 plants. If a 10% failure occurs, about 70 plants ready for planting will be obtained. In plantation with 2 × 3 m of spacing, only about 17 plantlets

per hectare are required. Thus, the in vitro multiplication technique of *L. suaveolens* propagation will be more efficient for plantation.

## 7. Conclusions

This research has a strategic value to obtain techniques for providing plantlets or seedlings on marginal lands that are difficult to manage. As an alternative to mass propagation of seedlings for peatland rehabilitation, we developed an in vitro multiplication technique of *L. suaveolens*, a water-resistant tree species. This study is a new high-efficiency innovation technique with a mutually reinforcing effect on seedling growth, increasing the multiplication coefficient in vitro by combining the PGRs BAP and NAA. The availability of balanced nutrients and PGRs during in vitro multiplication resulted in improved plantlets growth after acclimatization and ready-for-field planting. This protocol can be used to spread *L. suaveolens* on a large scale. This protocol will need to be tested on a larger scale of cultivation.

**Author Contributions:** A.I.P., N.K.K., A.N., S.S., A.R., L.H., T.H., P.B.S., R.S.W. and F.L.: contributed in conceptualization, methodology, software, validation, formal analysis, investigation, resources, data curation, writing—original draft preparation, writing—review and editing, visualization, supervision, and project administration. All authors had an equal role as main contributors in discussing the conceptual ideas and the outline, providing critical feedback for each section and writing the manuscript. All authors have read and agreed to the published version of the manuscript.

**Funding:** This research received no external funding.

**Institutional Review Board Statement:** Not applicable.

**Informed Consent Statement:** Not applicable.

**Data Availability Statement:** Not applicable.

**Acknowledgments:** The authors gratefully acknowledge the Research and Development of PT. Arara Abadi in Perawang, Riau, for providing the plant materials. We wish to thank to Ministry of Environment and Forestry of Indonesia for fully supporting this study.

**Conflicts of Interest:** The authors declare no conflict of interest; the funders had no role in the design of the study; in the collection, analyses, or interpretation of data; in the writing of the manuscript, or in the decision to publish the results.

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
