# Peer review of "In Vitro Multiplication of Lophostemon suaveolens (Sol.ex Gaertn.) Peter G.Wilson & J.T. Waterh): Peatland Tree Species for Rehabilitation"

_sustainability, doi:10.3390/su142214720_

Round 1

Reviewer 1 Report (Previous Reviewer 3)

Dear authors, this article “In Vitro Multiplication of Lophostemon suaveolens (Sol.ex Gaertn.) Peter G.Wilson & J.T. Waterh): Peatland Tree Species for Rehabilitations” (sustainability-1949294) demonstrated the effects of the in vitro multiplication of peatland species. The research demonstrated the potential production, yield, and benefits based on PGR (BAP and NAA) concentrations in growth and development.

Minor points:

-Alphabetic order keywords; and not repeat similar title words;

-Check all, references, style of citation and Model of the manuscript, following "Instruction of Authors"; In addition, Check all citation in manuscript, reference figures, formatation of subtitle (italics);

There is a scope for improvement in the discussion and conclusion section: a) additional emphasis on the significance of the study, b) scientific contribution of the paper; c) prospectively to other plants to Sustainability interest;

Check all legends in figures and x and y scientific notations.

-How this methodology can be extended for other plants: potential challenges, advantages? Add in your discussion topic, please.

-Please. All standardization of nomenclature equipment/reagents when necessary. Example: Fabricant, City, State, Country (three-letter). Check all manuscript.

Best Regards

Author Response

I heve corrected the revision from reviewers 1,2 and 3

Reviewer 2 Report (Previous Reviewer 2)

The manuscript is not suitable for publication. The experiment was very limited. Little results were obtained, and they raise my concerns. Many important methodological informations are missing.

Please read carefully my detailed comments and remarks in the attached pdf file.

Author Response

I have corrected the tevision from reviewers 1, 2 and 3

Reviewer 3 Report (Previous Reviewer 1)

The manuscript entitled In Vitro Multiplication of Lophostemon suaveolens (Sol.ex Gaertn.) Peter G.Wilson & J.T. Waterh): Peatland Tree Species  for Rehabilitations

after Minor Revision

In my opinion, the subject of this work is relevant for the Journal Sustainability MDPI after Minor revisions and some corrections.  

The topic of the paper is very interesting and important, especially because it is very rare and not enough investigated. The peatlands across the world present one

The Sustainability MDPI journal readers wants interesting and quality papers.

First, before all the paper has the next sections and sub-sections (i.e. Abstract, Materials and Methods, Multiplication of L. suaveolens, Rooting and acclimatization of L. suaveolens, Results, Multiplication of L. suaveolens, Rooting and acclimatization of L. suaveolens, Discussion, Implication, Conclusions, etc.).

The section of Abstract

In this section the authors must add minimum two sentences which explain the main results of this investigation and why and how they used NDVI (Normal Difference Vegetation Index) in hope to connect it with evapotranspiration.

The section Introduction

In this section the authors can only add more sentences about same or very similar results of peatlands restoration across the world and cite some of them.

Line 174, the Equation 2 must be better explained and statistical significant 0.05 better analyzed. The authors also must have answered on the questions?

1.      Why they used ANOVA test?

2.     Why they used Duncan's multiple range test (DMRT)?

Also the results from statistical analysis may be presented on graph or to make new Table.

The section Results

The Figure 1. All part of this Figure the authors must combine into one not separated.

The same for the Figure 4.

Figure 5. Please use the different colors (colors with strong contrast) for this Graph.

I strongly recommend to the authors add a new sub-section Limitation of research. In this sub-section, the authors must explain the strength and also the disadvantages of their research.

Section Conclusion

This section is overall short for this rank of Journal, please add more crucial sentences of this research in this section.

This paper has the potential to be published. The authors did a lot of things within this manuscript. The paper is very interesting and scientifically correct. Especially, in area of forest recovery.

In the end, I recommend Minor Revision.

Good luck to the authors 

The Reviewer#3

Author Response

I have corrected the revisions from reviewers 1, 2 and 3

Round 2

Reviewer 2 Report (Previous Reviewer 2)

I keep my decision form previous reviews. The reasons are the same, in general the scope of the experiment is very limited.

Author Response

Dear authors,

This research has not be able to fully reveal all the stages until obtaining mass plantlets for rehabilitation, further research is needed, however this is a very important initiation research

This manuscript is a resubmission of an earlier submission. The following is a list of the peer review reports and author responses from that submission.

Round 1

Reviewer 1 Report

The manuscript with the title Tissue Culture of Lophostemon sp: Peatland Trees for

Rehabilitations after Major Revision

In my opinion, the subject of this work is relevant for the Journal Sustainability after

approval of Major revision.

The topic of the paper is very interesting and important in the connection between

 Peatland Trees rehabilitation in Indonesia.

The journal readers of the Journal Sustainability seek and wants only quality papers.

First, before all, the structure of the paper is divided into the next sections and sub-sections (i.e. Abstract, Introduction, Materials and Methods, Multiplication in vitro of Lophostomon sp., Rooting in vitro and acclimatization of Lophostemon sp., Results, Multiplication of Lophostomon sp. in vitro, Rooting in vitro and acclimatization of Lophostemon sp., Discussion, Multiplication of Lophostomon sp. in vitro, Rooting in vitro and acclimatization of Lophostemon sp., Conclusions, implication).

In the section of Abstract, it is necessary to add sentences which explained better the methodology and the main results.

Generally, the text of this manuscript is very short and must be immediately extend following next way:

-The authors must compare all species of Lophostemon sp. in all habitats around of the world. This evergreen tree is very important for the flora and fauna, please explain how?

-Also for the potential readers it is good to know there were any similar action of plantation in Indonesia before? If yes, please the effects.

-Because it is very important for the saving of peatlands, the before investigation used the advanced techniques. I recommend to next papers for reading and citations:

Please cite these papers:

-Aleksandar Valjarević, Tatjana Djekić, Vladica Stevanović, Radomir Ivanović, Bojana Jandziković, GIS numerical and remote sensing analyses of forest changes in the Toplica region for the period of 1953–2013, Applied Geography, Volume 92,2018, Pages 131-139, https://doi.org/10.1016/j.apgeog.2018.01.016.

-Wenjiang Zhang, Qifeng Lu, Kechao Song, Guanghua Qin, Yan Wang, Xin Wang, Hongxia Li, Jun Li, Guodong Liu & Hua Li (2014) Remotely sensing the ecological influences of ditches in Zoige Peatland, eastern Tibetan Plateau, International Journal of Remote Sensing, 35:13, 5186-5197, DOI: 10.1080/01431161.2014.939779.

This section (Introduction) must be extend too. Because of importance of this research it would be fine to add position map of the researched area.

The section Materials and Methods

In this section the authors must better explained the Eq.1

It is the same for the Eq.2

According to these equations it is necessary to add some statistical analysis to find potential statistical importance. Or how these experiments work in practical sense, the authors must explain better.

Results

In this section the Figure 5, must be replaced by better Figures with better colors.

Figure 6, must be changed too. This Figure must be colorized in the way to be for potential readers more readable.

Discussion

This section present has the main problem because this manuscript in its present form can’t be accepted. In the section, the authors step by step must explain similar research and explain how can looks the future results.

Conclusion

This section is overall short and must be mandatorily extended. For this rank of Journal, it is necessary to write a concise Conclusion with the main results.

This manuscript of course deserves to be published, after Major revision. This manuscript has scientific potential and it describes rare subject and can be important for readership.

In the end, I recommend Major Revision.

Good luck to the authors

The Reviewer#1

Author Response

We will improve all reviewer directives

Reviewer 2 Report

I have read the manuscript „Tissue culture of Lophostemon sp: Peatland Trees for Rehabilitations” prepared by Asri Insiana Putri et al. for publication in Sustainability MDPI.

Generally the small scope of experiments, the description of the applied methodology, and the way, in which results are presented and interpreted,  are not applicable for publication. Some very important methodological informations are missing, and the interpretation of results seems to be very often incompatibile with the statistical analysis. The Introduction and Discussion chapters contain numerous deficiencies and inaccuracies. The terminology used in incorrect as for plant tissue cultures. The Authors did not provided full species name of tested plant. The quality of the manuscript is very low.

I am sending the pdf file with the manuscript, in which all my detailed comments and suggestions are written.

Author Response

Dear Reviewer

We have corrected the revision of the manuscript according to your direction

Reviewer 3 Report

Not recommended for publication. I suggest authors rewrite and re-edit manuscript sentences in introduction, M&M, results, discussion, implications, conclusions and check all-references. Best Regards

Round 2

Reviewer 1 Report

The manuscript with the title In Vitro Multiplication of Lophostemon suaveolens (Sol.ex Gaertn.Peter G.Wilson & J.T. Waterh): Peatland Tree Species for Rehabilitations

The authors answered to all of my comments and corrected all of mistakes within the text.

The Reviewer #3

Reviewer 2 Report

I do not recommend the publication of the revised manuscript in international journal with Impact Factor like Sustainability. The small scope of the experiment, the incorrect description and vocabulary of plant micropropagation, and poor English language are the most important reasons for my decision.